# Monitoring of Avian Influenza Viruses and Paramyxoviruses in Ponds of Moscow and the Moscow Region

**DOI:** 10.3390/v14122624

**Published:** 2022-11-24

**Authors:** Anastasia Treshchalina, Yulia Postnikova, Alexandra Gambaryan, Aydar Ishmukhametov, Alexei Prilipov, Galina Sadykova, Natalia Lomakina, Elizaveta Boravleva

**Affiliations:** 1M. P. Chumakov Federal Scientific Center for the Research and Development of Immune-and-Biological Products, Village of Institute of Poliomyelitis, Settlement “Moskovskiy”, 108819 Moscow, Russia; 2Department of Virology, Faculty of Biology, Lomonosov Moscow State University, 119991 Moscow, Russia; 3Gamaleya National Center of Epidemiology and Microbiology of the Russian Ministry of Health, 123098 Moscow, Russia

**Keywords:** avian influenza, diversity, circulation

## Abstract

The ponds of the Moscow region during the autumn migration of birds are a place with large concentrations of mallard ducks, which are the main hosts of avulaviruses (avian paramyxoviruses) and influenza A viruses (IAV). The purpose of this study was the determination of the biological diversity of IAV and avulaviruses isolated from mallards in Moscow’s ponds. A phylogenetic analysis of IAV was performed based on complete genome sequencing, and virus genomic reassortment in nature was studied. Almost all IAV genome segments clustered with apathogenic duck viruses according to phylogenetic analysis. The origin of the genes of Moscow isolates were different; some of them belong to European evolutionary branches, some to Asian ones. The majority of closely related viruses have been isolated in the Western Eurasian region. Much less frequently, closely related viruses have been isolated in Siberia, China, and Korea. The quantity and diversity of isolated viruses varied considerably depending on the year and have decreased since 2014, perhaps due to the increasing proportion of nesting and wintering ducks in Moscow.

## 1. Introduction

Avulaviruses, belonging to the *paramixoviridae* and Influenza A viruses (IAV), are enveloped negative-strand RNA viruses that cause disease in humans and domestic and wild animals [1,2,3]. The primary hosts of avulaviruses and avian influenza viruses (AIV) are wild waterfowl. Avulaviruses are phylogenetically divided into low virulent wild bird viruses and virulent poultry outbreak viruses [4,5]. Newcastle disease virus (NDV), which is caused by Avian orthoaviulavirus 1 (AOAV-1), has great economic importance [6]. Due to its global distribution, NDV is one of the most devastating infections in poultry [7].

Among AIVs, apathogenic viruses of wild birds and virulent viruses of other hosts also coexist. Natural hosts of AIVs are aquatic birds of the orders Anseriformes and Charadriiformes. AIVs with the H1–H12, H14, and H15 subtypes of hemagglutinin (HA), and with the nine subtypes of neuraminidase (NA), mainly circulate in ducks. Subtype H13 and H16 AIVs are usually found in gulls. The H5, H7, H6, and H9 subtypes are more common among poultry [8]. Pigs, horses, and humans are other potential hosts for influenza A viruses. In horses and pigs, the influenza virus usually causes nonfatal disease [9,10]. Nevertheless, swine IAV could be precursors of human virus, that pose a substantial threat in the potential initiation of new pandemic [11]. In 2021–2022, thousands of cases of highly pathogenic avian influenza (HPAI) were reported in Europe. HPAIs (H5) were detected in wild birds, which are associated with the risk of introducing the virus into poultry farms. Some of these viruses were also detected in wild mammalian species in the Netherlands, Slovenia, Finland, and Ireland, showing genetic markers of mammalian adaptation. There have been human cases of A(H5N1/N6) virus infection in the UK and China, and 15 cases of A(H9N2) virus infection in China and Cambodia [12].

Waterfowl play the main role in the circulation of AIVs and avulaviruses. Autumn migration of wild waterfowl contributes to the spread of their viruses [13]. During migration, most wild birds follow certain directions, the largest of which are designated as world bird migration routes. The habitats of water birds and flight paths provide independent circulation of viruses in the Eastern and Western hemispheres. The East Atlantic, Black Sea–Mediterranean, and East Asian–East African migratory routes of birds pass through the territory of the European part of Russia. The main nesting sites of many migratory birds of the Anseriformes order are located in the north of Russia [14]. The ducks nesting in Finland, Estonia, and northwestern Russia fly through Eastern Europe in autumn [15]. The spring migration of garganey *(Spatula querquedula)* from northern Nigeria to Russia has been described [16]. During the flight, migratory birds can be sources of infection for synanthropic species of birds and mammals [17].

The city ponds of Moscow and the Moscow region during the autumn migration of birds are the location of large concentrations of the mallard duck *(Anas platyrhynchos)*, which is one of the most numerous waterfowl species in Europe and the main carrier of the avian influenza viruses. In recent years, wild ducks increasingly do not fly away, but remain to winter in non-freezing urban water bodies [18]. The population of wintering wild ducks creates an additional method of virus spread, by contacting both wild and synanthropic birds [16].

The purpose of this study was to identify the biological diversity and ecological characteristics of AIVs and avulaviruses isolated from mallards in Moscow ponds. Based on complete genome sequencing, a phylogenetic analysis of AIVs was conducted. The natural reassortment and the virus pathways in the wild were investigated.

## 2. Materials and Methods

### 2.1. Reagents

Fetuin and horseradish peroxidase were from Serva, Oftringen, Switzerland. Antibodies against mouse immunoglobulins conjugated with horseradish peroxidase were from Sigma, St. Louis, MO, USA. Viral RNA Mini Kit was from QIAGEN, Hilden, Germany. MycoKill AB solution was from PAA Laboratories GmbH, Pasching, Austria. Ethidium Bromide Solution was from Promega, Madison, WI, USA. MMLV Reverse Transcription kit, random primers, nuclease free water, DNA Ladder, and TAE buffer were from Evrogen, Moscow, Russia. Ribonuclease Inhibitor was from Syntol, Moscow, Russia. Embryonated chicken eggs were purchased from the State poultry farm “Ptichnoe” (Moscow, Russia).

### 2.2. Viruses

Fresh feces of black-headed gulls *(Larus ridibundus)* and mallards *(Anas platyrhynchos)* were collected in 2006–2021 on the shore of city ponds in Moscow and Moscow region (see map for location of ponds in Appendix A). Feces were suspended in a double volume of phosphate-buffered saline (PBS), supplemented with 0.4 mg/mL gentamicin, 0.1 mg/mL kanamycin, 0.01 mg/mL nystatin, and 2% MycoKill AB solution. The suspension was centrifuged for 10 min at 4000 rpm, and 0.2 mL of the supernatant were inoculated into 10-day-old chicken embryos (CE). Allantoic fluid was collected after 48 h and tested by hemagglutination assay with chicken erythrocytes. Positive samples were taken for further passaging. All isolated AIV strains are stored in the virus repository of the Chumakov scientific center (Moscow, Russia).

### 2.3. Virus Typing

Hemagglutination-positive samples were tested in the fetuin-binding (FB) test with peroxidase-labelled fetuin [19] and in enzyme immunoassay with a mixed mice serum against AIVs of different subtypes. These reactions differentiated AIVs and avulaviruses. After that, positive samples were tested by PCR with influenza-specific primers, whereas negative samples were tested with the PMX primer set (PMX1 and PMX2) [20].

### 2.4. Sequencing

Viral RNA was isolated from the allantoic fluid of infected chicken embryos with a commercial QIAamp Viral RNA mini kit (Qiagen, Hilden, Germany, # 52904). Full-length viral genome segments were obtained by reverse transcription and PCR with specific terminal primers, MMLV, and Taq-polymerase (Alpha-Ferment Ltd., Moscow, Russia). The amplified fragments were separated by electrophoresis in 1–1.3% agarose gel and subsequently extracted from the gel with the Diatom DNA Elution kit (Isogene Laboratory Ltd., Moscow, Russia, # D1031). Sequencing reactions were performed with terminal or internal primers with the BrightDye ™ Terminator Cycle Sequencing Kit v3.1 (Nimagen, Nijmegen, The Netherlands), followed by analysis on an ABI PRISM 3100-Avant automated DNA sequencer (Applied Biosystems 3100-Avant Genetic Analyzer, Foster City, CA, USA). The Lasergene software package (DNASTAR Inc., Madison, WI, USA) was used for assembly and analysis of nucleotide sequences.

### 2.5. Downloading of Sequences and Evolutionary Tree Construction

The complete nucleotide sequences of AIV internal genes (PA, PB1, PB2, NP, MP, and NS) and external genes (H1, H3, H4, H5, N1, H6, H11, N1, N2, N3, N6, and N8) were downloaded from GISAID database (https://www.gisaid.org/ (accessed on 13 June 2022)) [21] and the Influenza research database (https://www.fludb.org (accessed on 13 June 2022)). The selected sequences were aligned by the MUSCLE method using the software package of MEGA X (https://www.megasoftware.net/ (accessed on 4 November 2020)) [22]. Time-scaled trees were generated for each internal segment with known isolation dates using BEAST [23] under GTR model with 1000 bootstrap replicates. The strict clock model of molecular clock was chosen for all segments.

## 3. Results

During the autumn periods of 2006–2021, one AIV virus strain was isolated from gull feces, and 57 AIV isolates and 28 avulaviruses were isolated from mallard feces. Representatives of AOAV-1 and Avian para-avulavirus 4 (APMV-4) were identified among the avulaviruses. Full names, designations of viruses, dates of isolation and GenBank accession numbers for sequenced AIV isolates are given in Appendix A.

### 3.1. Phylogenetic Analysis

Forty-six AIV isolates were fully sequenced. Nucleotide sequences were compared with AIV homologous sequences from the GISAID database [21]. The genome of the g/3100/06 (H6N2) virus was not separated from a number of duck viruses; that is, in this case, we can speak of a spillover of duck virus to gulls and not a specific gull virus. The genetic diversity of Influenza A viruses isolated at Moscow ponds is very high. For all genes, including hemagglutinins H1, H3, H4, H5, H6, and H11 and neuraminidases N1, N2, N3, N6, and N8, we found several evolutionary lineages that occurred in different combinations. Even when during one season the viruses practically coincided in terms of hemagglutinin and neuraminidase (as, for example, H4N6 viruses of 2009), some of their internal genes belonged to different clades (see Appendix A).

Viruses matching the clades of all genes were rare until 2015. We never detected viruses that coincided in clades of all genes in different years. In full accordance with the work [24], viruses circulated as a pool of genes that were intensively mixed and appeared annually in new combinations. Therefore, the study of the genetic relationships of the studied influenza viruses was carried out for each of the genes separately; the evolution of the virus as a whole could not be traced.

According to the PB2 gene, Moscow viruses can be classified into four groups. Most of the isolates are located on the evolutionary branch that includes subgroups of Asian and European viruses. This branch contains almost exclusively duck viruses. The PB2 gene of the g/3100/06 (H6N2) virus is located in a separate clade; it is closely related to the PB2 genes of the Swedish wild duck viruses (over 99% similarity). The PB2 of the d/3735/09 (H4N6) belongs to clade of Asian viruses, which included chicken H9N2 viruses. The d/5897/21 (H3N8) and d/5908/21 (H3N8) are closely related to poultry and wild bird H5 viruses. There are many poultry H7 viruses in the PB2 clade containing d/4971/13 (H5N3) and d/4788/12 (H3N8) Moscow isolates.

According to the PB1 gene, most viruses are also located on the broad evolutionary branch, which includes subclades of Asian and European duck viruses. Two viruses from 2008, d/3661/08 (H4N6) and d/3556/08 (H3N1), stand apart. Viruses d/4242/10 (H3N8), d/5586/18 (H1N2), d/5662/18 (H1N2), and d/5712/19 (H11N6) belong to a separate branch that includes duck viruses isolated in Europe, Asia, and Africa. This clade is abundant with H5 and H7 viruses. There are many chicken viruses in the clades containing d/4643/11 (H4N6) and d/5908/21 (H3N8) Moscow isolates.

According to the PA gene, d/4031/10 (H6N2) and d/4182/10 (H5N3) isolates are closely related to poultry H7 viruses. Four viruses from 2009–2010 belong to the European PA evolutionary lineage, which branched off into viruses of H13 and H16 subtypes.

The NP genes of Moscow isolates belong to ten clades. The most isolated European clade contains the only Moscow virus g/3100/06 (H6N2). This branch contained the Italian poultry H7N1 viruses (the NP gene of the g/3100/06 (H6N2) virus has 98.96% similarity with NP A/turkey/Italy/977/1999 (H7N1)). The second isolated clade contains the d/4031/10 (H6N2) virus. This clade mainly consists of Swedish viruses. A third small, isolated clade contains the d/3661/08 (H4N6) virus. Most of the viruses of this clade have European origin, although there are H9N2 isolates from India and China. At the end of the last century, this branch gave rise the NP of highly pathogenic chicken H9N2 and H5N1 viruses. Another clade included nine isolates from 2013–2019. This clade consists of Asian, European, and African isolates. In the early 2000s, the NP of this clade became part of a new genotype of highly pathogenic H5 viruses. The d/5897/21 (H3N8) and d/5908/21 (H3N8) isolates, as well as the d/5881/21 (H3N2) isolate, are closely related to H5 poultry viruses (similarity greater than 99%).

According to the M gene viruses, d/3556/08 (H3N1) and d/3720/09 (H6N2) belong to a branch that includes American, African, Asian, and European viruses (Figure 1). In addition to duck viruses, this branch contains viruses isolated from songbirds, ostriches, and guillemots (Order: Charadriiformes; Family: Auks). This evolutionary branch contains a large proportion of the chickens H7 subtype viruses.

The NS genes of the Moscow isolate AIVs belong to the A or B evolutionary lines. Nine isolates from the NS line B are evolutionarily close to the NS of poultry H7N1 viruses that caused the 1999 outbreak; the percentage of nucleotide similarity between the NS viruses d/3720/09 (H6N2) and A/turkey/Italy/977/1999 (H7N1) is 99.31%. Thirty-six isolates with NS line A are distributed over four evolutionary branches; the first contains European viruses and the other three contain Asian and European viruses. In all these four branches, there are practically no poultry viruses.

The HA H1 of d/4970/13 (H1N1) has 99.18% nucleotide similarity with A/Mallard/Sweden/816/2014 (H1N1) HA, and in accordance with [16], is similar to Asian viruses of previous years and to European viruses of subsequent years. The Moscow viruses of 2018 belong to the same clade, while two 2019 isolates are of Asian origin. The HA H3 genes are located on different halves of the Eurasian part of the H3 evolutionary tree. Isolates d/3806/09 (H3N8), d/5881/21 (H3N2), d/5908/21 (H3N8), and d/5897/21 (H3N8) belong to the Asian clade, with a small number of European viruses. The rest of the Moscow isolates belong to a large group of mostly European viruses, and of these, only the isolate d/5037/14 (H3N8) belongs to a mixed clade that includes both European and Asian viruses. The HA H4 genes of the Moscow isolates belong to four European duck AIV clades. The HA H5 genes of 2010 and 2013 belong to two branches of European AIV, which are not associated with the highly pathogenic H5 viruses. The HA H6 genes of the Moscow isolates belong to a large clade of European AIV, with single representatives from near Asia and Egypt. The HA H11 genes of two Moscow viruses are located in two different clades of the Eurasian AIV.

Neuraminidases N1 were found in Moscow isolates as part of the H3N1 and H1N1 viruses. NA of the d/4970/13 (H1N1) virus, likewise HA, has 99.51% similarity with the NA of A/Mallard/Sweden/816/2014 (H1N1) and is related to both Asian and European viruses. The isolates of 2019 also belong to the same clade. This clade contains some amount of chicken H5N1 viruses. NAs N2 in Moscow isolates were found in H6N2, H3N2, and H1N2 viruses. NA of the d/3720/09 (H1N1) virus belongs to a clade including poultry H5 and H9 viruses. NAs N3 were found only in H5N3 viruses. Just like the H5 genes, they belong to two independent branches of the European AIV. The clade with d/4971/13 (H5N3) и d/4952/13 (H5N3) isolates contains poultry viruses. NAs N6 were present in all viruses with H4 HA, as well as in H3N6 and H11N6 viruses. All of them belonged to European clades. The N6 of the d/5712/19 (H11N6) virus is close to the N6 of a number of viruses from 2011 and 2012, while the N6 from the viruses of 2008, 2009, 2010, and some of the viruses from 2011 are evolutionarily distant from each other. In H4N6 viruses, grouping of N6 by clades coincide with grouping of H4. NAs N8 were present in almost all H3 HA viruses. N8 of viruses d/4242/10 (H3N8) and d/5037/14 (H3N8) stand apart from other viruses. N8 viruses from 2010 were divided into two groups, repeating the division of H3. Similarly, the d/4661/11 (H3N8) virus is separate from other H3N8 viruses of this year, by both NA and HA. In the remaining H3N8 viruses, NA and HA are grouped by year. NA N9 was present in a single isolate d/3641/08 (H11N9), which belonged to a separated European group of H11N9 viruses.

### 3.2. Natural Reassortment of Gene Segments

The reassortment of genomes is well visible in the Moscow isolates from 2011–2012 (Figure 2). In 2011, eight strains were isolated at pond#1. Three of them (d/4494/11 (H3N8), d/4518/11 (H4N6) and d/4661/11 (H3N8)) did not match the clades of any of the genes. Isolate d/4528/11 (H4N6) contained the fourth variant of all genes, except for NP, which almost coincided with the NP of virus d/4494/11 (H3N8). Viruses d/4528/11 (H4N6) and d/4641/11 (H4N6) were practically twins—four of their genes coincided completely, and four others had minimal differences. Probably, the second of them, isolated two weeks after the first, is its direct descendant. Mix-strain d/4524/11 (mix) contained a mixture of neuraminidase N2 and N8. In six genes, this strain is very close to strain d/4494/11 (H3N8), but differs in the PB1 gene, as well as in both neuraminidases, since N8 neuraminidases of d/4524/11 (mix) and d/4494/11 (H3N8) have large differences. Strain d/4652/11 (H4N6) is the result of multiple reassortments. Three of its genes (PB2, PB1, and NP) are very similar to strain d/4494/11 (H3N8); HA and NA almost coincide with HA and NA d/4528/11 (H4N6), PA is a direct descendant of PA d/4518/11 (H4N6), and the M and NS genes are of independent origin. The d/4643/11 (H4N6) strain inherited the HA, NP, NA, and NS genes from d/4518/11 (H4N6); gene M from d/4661/11 (H3N8) and the genes of three polymerases are of independent origin. Strain d/4681/11 (H3N8), the only one isolated from pond#2 in 2011, coincides with all genes with d/4494/11 (H3N8), being, apparently, its direct descendant. It was isolated two months later in another pond, where it may have been the only AIV variant in this season, which ruled out natural reassortment.

The direct descendants of the AIV genes from 2011 were also found in 2012 on pond#2 and pond#3. Three strains (d/4771/12 (H4N6), d/4843/12 (H4N6), and d/4781/12 (H4N6)) inherited the HA and NA genes from d/4528/11 (H4N6). The same strains plus d/4788/12 (H3N8) inherited the PB2 and/or M and/or NS genes in various combinations from strain d/4494/11 (H3N8). The very high similarity proves the direct origin of these 2012 genes from the 2011 genes. The remaining genes of the 2012 isolates are independent of the 2011 isolates.

### 3.3. Dynamics of Virus Isolation by Years

The number and composition of isolated viruses varied dramatically over the years (Table 1). For example, in 2011, out of 385 samples collected, 18 AIVs and 13 avulaviruses were isolated, while in 2016, out of 115 samples, only one avulavirus, and in 2018, out of 494 samples, only two H1N2 influenza viruses were isolated. Until 2014, several subtypes of AIV were isolated in each season, and strain matches for all genes were rare, whereas in 2015, 2018, 2019, and 2021, several viruses were isolated that were almost identical in all genes (2–5 differences for the entire genome). That is, there was a decrease in the prevalence of viruses, and a decrease in the diversity of isolated viruses.

### 3.4. Circulation of Influenza Viruses between Moscow and Other Regions

Moscow is located in the overlap zone of the East Atlantic and Black Sea–Mediterranean bird migration routes. In order to identify areas with which Moscow has an intensive exchange of avian influenza viruses and to assess possible routes of transmission of the virus by wild waterfowl, a BLAST analysis was performed based on the GISAID database. For each gene of each of the viruses, the most closely related variant of non-Moscow origin was found. The percentage of nucleotide differences from the Moscow isolate is shown in Figure 3. If the difference exceeded 0.5% for the M and NS segments or 1% for the remaining segments, then these values are not shown. We chose these values because the M and NS segments are conserved in avian influenza viruses [25].

The vast majority of closely related viruses have been isolated in the Western Eurasian region, in particular the Netherlands, Sweden, Belgium, Ukraine, Georgia, and Italy. Much less frequently, closely related viruses have been isolated in Siberia, China, and Korea. However, there were no viruses with all segments of the genome of Asian origin, which may indicate that there is no direct transfer of viruses from Eastern Eurasia to Moscow.

We noted a decrease in the number of viruses with a high percentage of similarity after 2014. To test this hypothesis, we counted the number of viruses from regions other than Moscow that have a percentage of nucleotide differences of less than 0.5% for the M and NS segments and less than 1% for the remaining segments in comparison with each of the Moscow viruses. The results of the analysis are presented in Table 2. Viruses isolated in the European part of Eurasia and in countries bordering the Black and Caspian Seas were noted as Western Eurasian. The rest of the countries and part of Russia beyond the Ural Mountains were classified as Eastern Eurasian.

For all genome segments, the number of closely related viruses from Western Eurasia has been significantly decreasing since 2014. Until 2014, Moscow isolates, according to all genes, were closely related to a large number of strains in Europe and the Caucasus, i.e., there was an intensive circulation of viruses between these regions and Moscow. Since 2015, the circulation between Moscow and Western Eurasia has been sharply reduced. On the other hand, the virus exchange with Eastern Eurasia, which was not high before, has practically not changed since 2014. The only exception is the of enhancing the number of closely related viruses for the M gene. That enhancing is due to the presence of more than 30 viruses related by the M gene to the Moscow isolate d/5586/18 (H1N2). That may be related to the high conservatism of the M gene in birds, because of which, viruses of several years of isolation fall into the sample with more than 99.5% similarity [25].

## 4. Discussion

Since 2006, we have been studying the diversity of AIVs, AOAV-1, and APMV-4 carried by wild waterfowl in five ponds in Moscow and the Moscow region.

Viruses have been shown to move freely from pond to pond, following the movements of ducks. Viruses that were very similar to the Moscow isolates of the previous year in a number of genes have been repeatedly isolated. The presence of a very few nucleotide substitutions may indicate that the virus did not multiply in living organisms most of the time during the year. We assume that the viruses could survive in a pond that freezes for the winter until the next season [26].

Phylogenetic analysis showed that almost all of the genes of the studied viruses belong to the pool of apathogenic duck viruses; although some of them are located on the branches of evolutionary trees that gave rise to chicken viruses in the past.

The majority of Moscow isolates belong to clades including both European and Asian viruses, although there are a few exceptions. For example, the NP and M genome segments of A/duck/Moscow/4031 virus belongs to strictly European clades.

All Moscow isolates are harmless for mice and chickens [27]. Nevertheless, some of their genome segments are closely related to segments of the highly pathogenic chicken viruses of H7 and H5 subtypes, as well as H9N2 chicken viruses. Thus, the PB2, PB1, and NP genes of the d/5908/21 (H3N8) and d/5897/21 (H3N8) viruses have above 99% similarity to the corresponding genes of poultry H5 viruses.

Until 2014, many viruses from other places closely related to the Moscow isolates were detected. The origin of the genes of Moscow isolates were different; some of them belong to European evolutionary branches, some to Asian ones. In the latter case, in addition to closely related Asian viruses, a certain number of closely related European viruses were found in the databases. The analysis showed that the most intensive circulation of viruses occurs between Moscow and the countries of Europe and the Caucasus, and, to a lesser extent, the Far East and Siberia. One case was noted where the closest relative was a virus from Egypt. The coast of the Nile and the Mediterranean Sea are among the main places where mallard ducks winter, so the isolation of a virus closely related to the Moscow virus in Egypt is quite natural. Probably, the path of Asian viruses to Europe and Moscow goes through Africa, where the AIVs transmit and reassort in the wintering birds.

After 2014, the virus isolation pattern changed. Viruses became isolated less often, and the diversity of isolated viruses fell. The number of closely related viruses from Western Eurasia has fallen sharply, while the exchange of viruses with the countries of Eastern Eurasia has remained at the same level. This probably reflects changes in duck migration routes.

In 2016, the highly pathogenic Asian viruses of the H5 subtype appeared in Europe [28]. Since then, outbreaks of H5 in Europe and Russia have repeated regularly. It is possible that some countries are pursuing a policy to reduce the population of migratory ducks, which are the main carriers of these viruses. In addition, both in Russia and in Europe, more and more often ducks do not fly away in autumn but remain to winter on non-freezing reservoirs and are fed by citizens. This, too, could reduce the transit of European viruses into Moscow.

## 5. Conclusions

In 2006–2021, avian influenza and paramyxoviruses were monitored in wild ducks during an autumn migration through Moscow. Fifty-eight AIV strains and 28 avulaviruses strains were isolated. Viruses of both European and Asian origin were isolated; however, the immediate predecessors of Moscow viruses, as a rule, were European viruses. In recent years, the circulation of AIVs between Moscow and Europe has decreased, although the circulation with East Asia has remained at the same level. This probably reflects changes in duck migration routes.

## Figures and Tables

**Figure 1 viruses-14-02624-f001:**
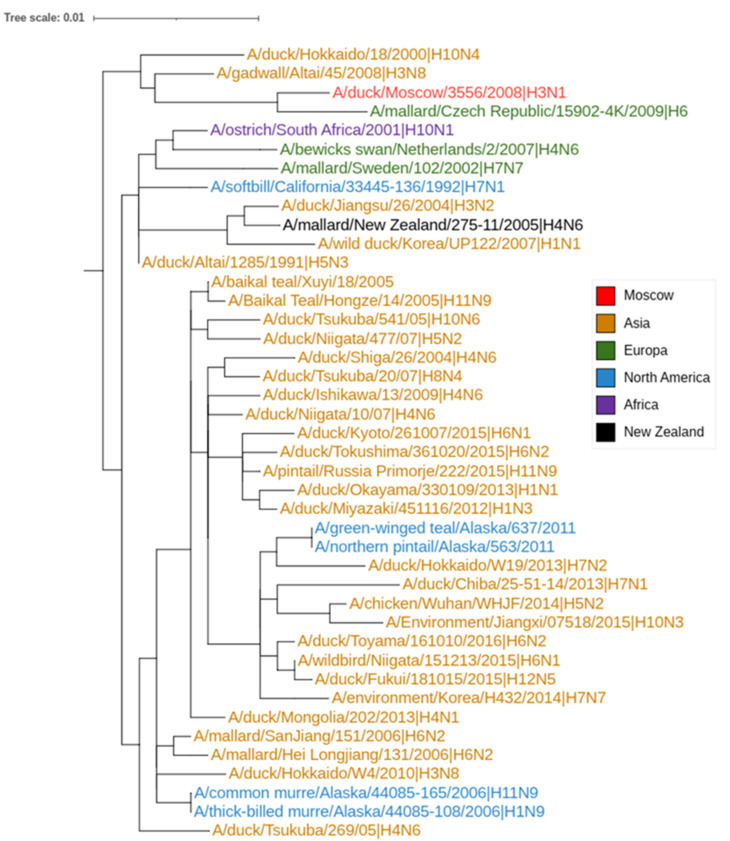
Fragment of the M gene phylogenetic tree. The color shows the geographical origin of the viruses.

**Figure 2 viruses-14-02624-f002:**
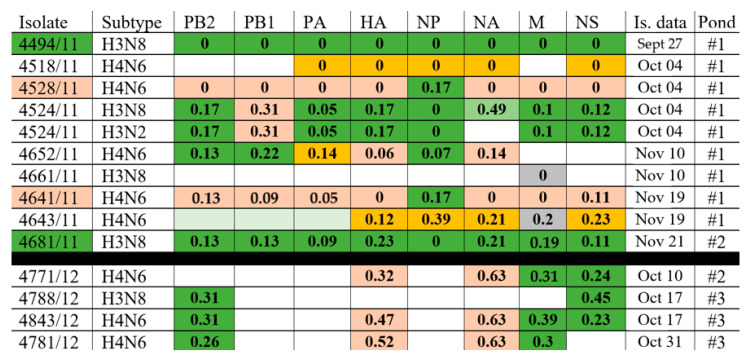
Examples of natural reassortment in 2011 and 2012 isolates. Gene variants that differ by more than 1% of nucleotide substitutions are marked in different colors. The numbers in the cells reflect the % of nucleotide substitutions relative to the origin isolate (cells are marked in the same color and with the number 0). The blank cells reflect the more than 1% of nucleotide substitutions relative to all the isolates of 2011.

**Figure 3 viruses-14-02624-f003:**
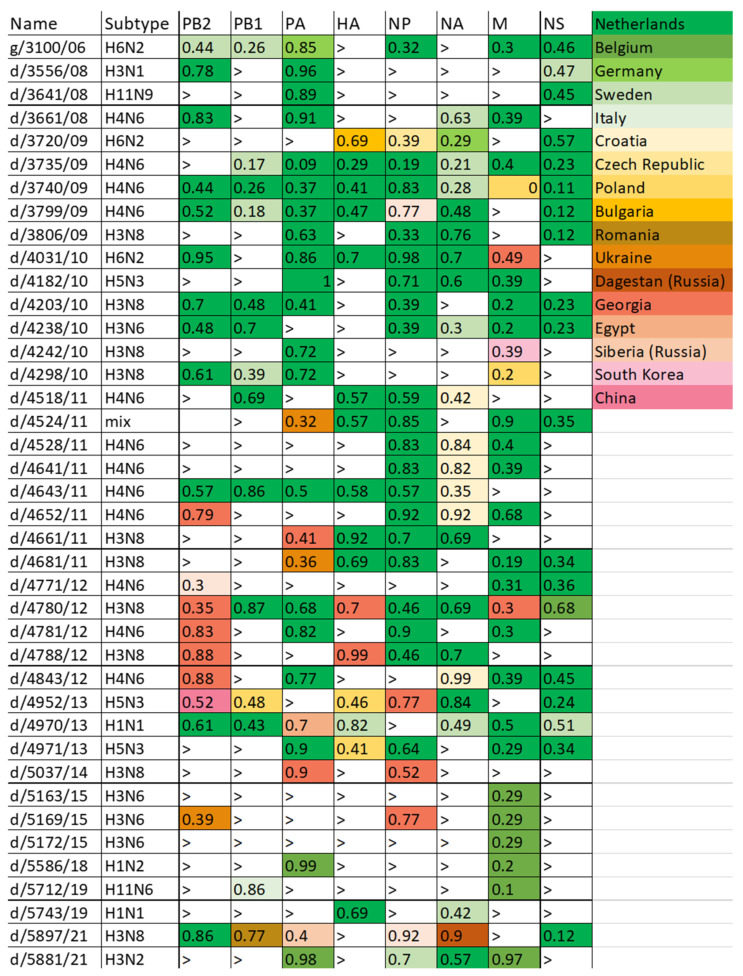
Regions of isolation of the most closely related virus for Moscow isolates. The color of the cells corresponds to the region of isolation of the most closely related variant relative to the Moscow isolate. The numbers in the cells reflect the % of nucleotide substitutions relative to the Moscow isolate. The > sign in the cells means that the difference exceeded 0.5% for the M and NS segments or 1% for the remaining segments.

**Table 1 viruses-14-02624-t001:** Isolation of avulaviruses and influenza A viruses from bird feces collected from the ponds in the Moscow region.

Years	Number Samples	Number Isolates Avulaviruses	Subtypes Avulaviruses ^1^	Number Isolates AIV	Subtypes AIV ^1^
2006	5	0		1	H6N2 (1)
2007	25	0		0	
2008	193	9	AOAV-1 (5)	4	H3N1 (2)
			APMV-4 (2)		H4N6 (1)
			Avulavirus n.s. (2)		H11N9 (1)
2009	151	0		5	H4N6 (3)
					H3N8 (1)
					H6N2 (1)
2010	441	3	Avulavirus n.s. (3)	7	H3N8 (3)
					H3N6 (1)
					H5N3 (2)
					H6N2 (1)
2011	385	13	AOAV-1 (6)	18	H4N6 (7)
			APMV-4 (4)		H3N8 (5)
			Avulavirus n.s. (3)		H3 mix (1)
2012	198	0		7	H4N6 (5)
					H3N8 (2)
2013	126	1	Avulavirus n.s. (1)	3	H5N3 (2)
					H1N1 (1)
2014	184	0		1	H3N8 (1)
2015	106	0		4	H3N6 (4)
2016	115	1	Avulavirus n.s. (1)	0	
2018	494	0		2	H1N2 (2)
2019	341	1	Avulavirus n.s. (1)	3	H1N1 (2)
					H11N6 (1)
2021	192	0		3	H3N2 (1) H3N8 (2)

^1^ In parentheses, the number of isolates of a given subtype.

**Table 2 viruses-14-02624-t002:** Average number of viruses from Western and Eastern Eurasia closely related to each of the Moscow isolates for 2006–2014 and 2015–2021.

	Years	PB2	PB1	PA	HA	NP	NA	M	NS
Western Eurasia	2006–2014	3.61	3.15	4.48	3.67	6.15	4.55	5.21	2.88
	2015–2021	0.17	0.33	0.06	0.17	0.25	0.67	1.50	0.08
	*p*-value *	<0.01	<0.01	<0.001	<0.01	<0.001	<0.01	<0.01	<0.001
Eastern Eurasia	2006–2014	1.36	0.00	0.00	0.00	0.18	0.06	0.15	0.00
	2015–2021	0,00	0.08	0.17	0.00	0.25	0.17	4.58	0.08
	*p*-value	>0.05	>0.05	>0.05	>0.05	>0.05	>0.05	<0.05	>0.05

* The level of significance of differences between different time intervals was calculated based on Student’s *t*-test.

## Data Availability

Not applicable.

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
