# Peer review of "Monitoring of Avian Influenza Viruses and Paramyxoviruses in Ponds of Moscow and the Moscow Region"

_viruses, 2022, doi:10.3390/v14122624_

Round 1

Reviewer 1 Report

In this manuscript, 46 AIV isloates from Moscow ponds were sequenced, compared withe other AIV sequences from the GISAID database and natural reassortment analysis. Europe-origin AIVs has decreased and East Asia-origin AIVs has increased. Ii indicated that the wild waterfowls migration might change and it might accelerate the ressortion of AIV in different geographical regions. 

My comments to the authors are as follows (minor points):

1. The sampled ponds should be marked on the map.

2. AMPV isolates is not neccessary for this manuscript.

3. The results showed that most of the isolates might be pathogenic to ducks, and whether there were abnormal feces or dead waterfowl.

Please check for spelling mistakes, eg. section 2.4 reverse transcription.

Author Response

Dear reviewer!

Thank you for your useful comments on our manuscript “viruses-2022296”

As suggested, we have attempted to improve the clarity of data presentation.

  1. The sampled ponds should be marked on the map.

           Map with ponds` location is included to supplemental material

  1. AMPV isolates is not neccessary for this manuscript.

           Incorrect taxonomy was fixed

  1. The results showed that most of the isolates might be pathogenic to ducks, and whether there were abnormal feces or dead waterfowl.

           The pathogenicity of the isolates was evaluated and described in “Boravleva E, Treshchalina A, Postnikova Y, Gambaryan A, Belyakova A, Sadykova G, Prilipov A, Lomakina N, Ishmukhametov A. Molecular Characteristics, Receptor Specificity, and Pathogenicity of Avian Influenza Viruses Isolated from Wild Ducks in Russia. Int J Mol Sci. 2022 Sep 16;23(18):10829”.

Please check for spelling mistakes, eg. section 2.4 reverse transcription.

Done

Reviewer 2 Report

Manuscript ID Viren-2022296 entitled "Surveillance of Avian Influenza Viruses and Paramyxoviruses in Ponds of Moscow and the Moscow Region" provides molecular biological results and phylogenetic analyzes of Paramyxoviridea and Influenza A viruses from fecal samples of birds from the Moscow area, Russia from 2006 to 2021. The topic is important as avian flu is a major disease problem in Europe in 2021/22. The used methods for virus analysis seem adequate, but I'm missing some results.

The entire manuscript should be revised to use the current taxonomy of both viruses, esp. avian orthoavulavirus 1 as species name with strains Newcastle disease virus and Pigeon paramyxovirus 1, Influenza A virus as species name.

Line 43/44: This part is incorrect because the influenza A virus of subtype H1N1 was an important cause of the human influenza pandemic at the beginning of the last century. This part needs to be rewritten as influenza is one of the most important viral diseases with the potential for pandemic spread.

Line 61: Add the taxonomic name in parentheses after the common name.

Line 65: Add the taxonomic name in parentheses after the common name and delete the taxonomic name on line 124.

Line 74: Please indicate the number and location of ponds (distance of ponds from each other) as well as the number of fecal samples in total and per year and the method of fecal sampling.

Lines 124-125 are part of the Materials and Methods section. Please explain how you can ensure that you only collected droppings from mallards and black-headed gulls. Taxonomic names must be written in italics.

Line 131 and 135: What is the reason only 46 of 57 AIV isolates were sequenced?

Line 131: You isolated 28 Paramyxoviridea but only listed 10 in Table 1, what happened to the other isolates?

Author Response

Dear reviewer!

Thank you for your useful comments on our manuscript “viruses-2022296”

As suggested, we have attempted to improve the clarity of data presentation.

The entire manuscript should be revised to use the current taxonomy of both viruses, esp. avian orthoavulavirus 1 as species name with strains Newcastle disease virus and Pigeon paramyxovirus 1, Influenza A virus as species name.

           Incorrect taxonomy was fixed

Line 43/44: This part is incorrect because the influenza A virus of subtype H1N1 was an important cause of the human influenza pandemic at the beginning of the last century. This part needs to be rewritten

           This section was rewritten

Line 61: Add the taxonomic name in parentheses after the common name.

           Done

Line 65: Add the taxonomic name in parentheses after the common name and delete the taxonomic name on line 124.

           Done

Line 74: Please indicate the number and location of ponds (distance of ponds from each other) as well as the number of fecal samples in total and per year and the method of fecal sampling.

           Map with ponds` location is included to supplemental material

Lines 124-125 are part of the Materials and Methods section. Please explain how you can ensure that you only collected droppings from mallards and black-headed gulls. Taxonomic names must be written in italics.

           Paragraph after line 124 has been moved to "Materials and Method” section.

           Taxonomic names was fixed.

Line 131: You isolated 28 Paramyxoviridea but only listed 10 in Table 1, what happened to the other isolates?

            Corrections made to the table

Round 2

Reviewer 2 Report

Thank you for editing your manuscript. All raised points have been fully answered.